# (Finite-Time) Thermodynamics, Hyperbolicity, Lorentz Invariance: Study of an Example

**DOI:** 10.3390/e27070700

**Published:** 2025-06-29

**Authors:** Bernard Guy

**Affiliations:** Mines Saint-Etienne, Institut Mines Télécom, 42100 Saint-Étienne, France; bernard.guy15@wanadoo.fr

**Keywords:** finite-time thermodynamics, finite resources, Lorentz invariance, entropy production, optimization, complementarity relations, hyperbolic problem, conservation of matter, four vectors, non-equilibrium

## Abstract

Our study lies at the intersection of three fields: finite-time thermodynamics, relativity theory, and the theory of hyperbolic conservation laws. Each of these fields has its own requirements and richness, and in order to link them together as effectively as possible, we have simplified each one, reducing it to its fundamental principles. The example chosen concerns the propagation of chemical changes in a very large reactor, as found in geology. We ask ourselves two sets of questions: (1) How do the finiteness of propagation speeds modeled by hyperbolic problems (diffusion is neglected) and the finiteness of the time allocated to transformations interact? (2) How do the finiteness of time and that of resources interact? The similarity in the behavior of the pairs of variables (x, t and resources, resource flows) in Lorentz relativistic transformations allows us to put them on the same level and propose complementary-type relationships between the two classes of finiteness. If times are finite, so are resources, which can be neither zero nor infinite. In hyperbolic problems, a condition is necessary to select solutions with a physical sense among the multiplicity of weak solutions: this is given by the entropy production, which is Lorentz invariant (and not entropy alone).

## 1. Introduction

“Finite-time thermodynamics” studies the evolution of non-equilibrium systems subject to finiteness constraints: finite resources, finite time allocated to transformations, etc. A criterion is used to define the values of physical quantities that make a given objective optimal (see, among many references, [1,2,3,4,5,6,7]).

It should be emphasized that the established term “finite-time thermodynamics” requires the incorporation of a minimum dissipation criterion. We put (finite-time) in parentheses if we emphasize the first part (finite-time) at the expense of the second (minimum dissipation), but, for us, the second part is indeed present.

Studies on the subject are generally based on a single space-time reference frame. They do not consider the case of another reference frame moving relative to the first. A relativistic approach raises a number of questions: How do finiteness constraints change? How do quantities change and accommodate the contraction/dilation effects established for space and time? How do optimization criteria change? What compatibility is there between different points of view?

We will not address these questions in general terms but rather in a specific manner, shifting them in two new directions that constitute the two objectives of our research:

(1) *How does thermodynamics in finite time work for large systems in terms of spatial dimensions?* We are thinking of geological systems measuring tens or hundreds of meters, or even more. In mathematical models, the transport of constituents within such systems is not driven by diffusion and its infinite propagation speeds but by advection with finite propagation speeds (diffusion is negligible at the scale considered). It is, therefore, appropriate to consider weak solutions in the mathematical sense for the propagation equations, simulating the displacement of discontinuities (hyperbolic problems). The finiteness of the velocities will lead to the finiteness of the times through simple relationships. The word geology attached to the example we will discuss simply indicates that we are dealing with large scales of space and time, where propagations, regardless of their actual speeds, have time to become noticeable.

(2) *What is the link between the finite space and time of (finite-time) thermodynamics and the finite resources it discusses?* Indeed, thermodynamics in finite time encompasses studies in which finiteness concerns both time and resources using the same methods. Does this not point us towards a deeper organic link expressing a form of equivalence between time and resources? Addressing this question brings us back to the fundamental philosophical question of the nature and existence of time. We have addressed this question in our works using relational logic: time and space have no definition in themselves; they are names for relationships between phenomena, or in this case, between resources. This is one way of interpreting the theory of relativity, which is expressed mathematically in Lorentz’s relations. The pairs of quantities (x, t and resources, resource flows) behave in the same way with respect to Lorentz transformations. The mathematical development proposed in this study will, thus, lead us to propose organic relationships between time and finite resources.

These two directions are, a priori, independent of each other. Their possible intersections constitute complementary research directions that will be outlined in the conclusions of this article.

Addressing these research directions requires articulating three fields: thermodynamics, in particular finite-time thermodynamics; the theory of relativity; and the theory of hyperbolic conservation laws. Each of these fields is very rich, and we will not go into their details here. Therefore, we will present a very simplified view of each of them, focusing on their philosophical foundations (that have been renewed to a certain extent) without exploiting their full richness.

Thermodynamics, in its elementary structure, studies evolving systems (second law) subject to conservation constraints (first law). Similarly, thermodynamics in finite time is understood here in its simplest form, based on the very words that define it: finite time, i.e., the study of systems for which a finite evolution time is considered, whether imposed or derived from the solution of the problem. In this context, the question of temperature, although essential in finite-time thermodynamics, will not be addressed directly, although it is hidden in entropy, and entropy production will be the dissipation criterion to be minimized. The duality of conservation and evolution, which is at the heart of thermodynamics, refers to the duality of space and time discussed in the theory of relativity. We look at the latter theory with a philosophical perspective (see below) that is also fruitful in thermodynamics.

For us, the theory of relativity is fundamentally a theory that proposes to construct and discuss the concepts of space and time by choosing a standard of motion that allows different systems in relative motion to interrelate. This interpretation, which is an extension of classical interpretations, is set out in our numerous works. It is based on a relational logic in which objects of thought (space, time) are the names of comparisons between material entities, which we will refer to here as the (finite) resources studied, as we have said. It is not so much the high value of the speed of light that matters as the relationships between the different speeds involved (in our relational understanding, important in relativity are the v/c ratios, whatever the value of the standard c, as if it could be considered on its own, which cannot be done in general terms: for us, the expression “speed of light” has no meaning by itself).

To gain this broader view of relativity, we draw on philosophy. The strength of physics lies in giving full value to the tools it uses (measurements, concepts, mathematical symbols, equations). It assumes correspondence with the compartments of reality that are supposed to exist separately (we can speak of realism or, in a slightly different sense, positivism or pragmatism). This allows for remarkable efficiency. Philosophy, on the other hand, confronts an infinitely complex reality where everything is connected to everything else. The world infinitely exceeds all representations we can make of it. Philosophy never stops reflecting on the conditions that make objective knowledge possible and on the paths that lead us to adopt one or another concept of physics. There are moments in the history of physics when it is essential to take a philosophical step back from what has been constructed, in order to possibly shift our choices in understanding reality. This, therefore, concerns relativity but, also, in another way, thermodynamics, as we have shown in our works. This allows us to accommodate various difficulties (conceptual, experimental, mathematical) that have arisen since the last construction. But efficiency in our actions will require a new pause in fundamental questioning.

Finally, the theory of hyperbolic problems is provided by the example chosen. This field is not well known to thermodynamicists (it is present in its own way in extended irreversible thermodynamics [8,9]; see also [10]). The term “hyperbolic problem” refers to the classification of partial differential equations (parabolic, elliptic, hyperbolic). A hyperbolic problem exhibits solutions with finite propagation speeds. It gives rise to discontinuous solutions and must, therefore, be placed in an appropriate mathematical framework, that of distributions. We then face a multiplicity of solutions (known as weak solutions), and a criterion must be added to select those that make physical sense. This is entropy production, and this structure (equation + selection criterion), thus, *shows a similarity to finite-time thermodynamics*. The fact that the example chosen comes from geology is not important. What is important is the mathematical structure of the hyperbolic problem, its concern with discussing different contrasting scales of space and time, and its ability to deal with real discontinuities in the sense of distribution theory.

In return for these simplified/modified views on thermodynamics, relativity, and the theory of hyperbolic conservation laws, we will provide original results, instructive answers to the two specific questions specified above, and research proposals for the scientific community. The logical structure of our approach is summarized in Figure 1.

Four boxes are distinguished, as variants of thermodynamics in finite time in the broad sense. We make a distinction, along the vertical axis, between the treatment of continuous physical quantities (bottom) and discrete quantities (top); along the horizontal axis, between the finiteness of time in the strict sense (left) and the finiteness of resources (right). The transition from the lower level to the upper level corresponds to larger scales of space and time where propagation speeds are finite and diffusion is neglected (we mark this with *D* → 0; discontinuous solutions; hyperbolicity of the problem). The transition from the left level to the right level amounts to exchanging the reference parameters, i.e., time and space (left) and flow of resources (right), according to a relational interpretation of the theory of relativity (they behave in the same way in Lorentz transformations). Arrows 1 and 2 refer to topics discussed in this study, as mentioned above, while arrows 3 and 4 represent research directions briefly discussed in the conclusions.

Our plan will be as follows. In the first part (theoretical background), we will set out the useful formulations for placing ourselves in a relativistic framework (in particular, using four vectors, writing the transformation of quantities and requesting Lorentz invariance). In the second part (results), we present the example studied and its solution: the finiteness relations express the conservation of matter, and the criterion for optimizing or choosing physical quantities is the production of entropy. We will look at how the various quantities and equations used (conservation, optimization criterion) are transformed. We will highlight the complementary-type relationships between time finiteness and resource finiteness. We conclude with a discussion section.

## 2. Theoretical Background

### 2.1. Revision of Mathematical Tools

If we place ourselves in a framework compatible with the relativist approach, we must revise our conceptual equipment. It is useful to deal with physical quantities not “on their own” but by associating them in pairs that allow us to define four vectors: it is these that will be the subject of relativistic transformations. The pairs (position, time; energy, momentum; electric charge, current) are among the best known. In relativity, we extend this formalism to three-component vector pairs such as electric field, magnetic field in appropriate tensorial mathematical entities.

In the present thermodynamic approach, we will have to deal with entropy in this way. What pair should we consider for it? For reasons that will become clear later, we propose the pair (entropy, entropy flow), denoted (*S*, *F*), where entropy *S* is a scalar depending on the variables *f* of the system, and *F* is a three-dimensional vector. This pair is already used as such in the solution of the hyperbolic-type equations we will be dealing with in our example (part 3) Entropy is a non-conserved quantity but this does not affect our choice. In the Lorentz transformation (see below), we will retain the entropy production associating the two vectors S and F (in their temporal and spatial derivatives, respectively): this mathematical form is conserved regardless of its sign. As we will see it is remarkable that Kruzhkov’s formulation of hyperbolic problems puts equalities and inequalities on the same footing. It is possible to choose another time reference based on the pair (f, g) that satisfies the entropy condition (Section 2.4), and this will not change the behavior with respect to Lorentz invariance. It is tantamount to placing ourselves in a potentially open system, which is in line with the conclusions of quantum thermodynamics: there is no rigorously closed physical system that has no relationship with the rest of the universe [11]. There is still controversy surrounding the choice of an entropic optimization criterion in relativistic thermodynamics, as we will discuss further below.

For the concentration variables used in our example, we must similarly consider a pair (concentration, concentration flux) for a small, open rock system; we will call it (*f*, *g*).

### 2.2. Links Between the Concepts of Space and Time

Our work on relativity has led us to renew our interpretation of the concepts of space and time (see, for example, [12,13,14]). Our approach is not one of substantial epistemology, in which the concepts of space and time are each grasped for their own sake; it is of relational epistemology, in which they are grasped in opposition/composition to each other, based on a comparison between phenomena or movements (these being envisaged in a primary way before words, through experience, experimental protocols, embodied cognition, designation). To space correspond the relative movements that are barely perceptible or “stopped”, compared to faster movements on which time is defined. Quantitative measurements are made by giving a particular movement the role of a standard: this is the meaning of the second postulate of the theory of relativity. In this context, it is interesting to take up the mathematization of time using three intermediate parameters: the coordinates of the position of the mobile that defines time (the photon in the atomic clock has replaced the relative position of the sun and the earth), i.e., *t_x_*, *t_y_*, *t_z_*. From these three coordinates, we can define the scalar *t* in various ways, for example, by $t=\sqrt{t_{x}^{2}+t_{y}^{2}+t_{z}^{2}}$. This formula is a way of constructing the usual time as a scalar. The invariance of the relativistic distance is another topic; it may also be written with the three intermediate pre-time coordinates. This approach offers greater symmetry between spatial and temporal variables and opens up new possibilities compared with the standard approach. For entropy, we need to define an intermediate vector with three components, *S_x_*, *S_y_*, *S_z_*, measuring the inhomogeneity of a system along the three directions of space and enabling us to define the entropy scalar. This is useful for generalizing the results that follow, but, no more than for time, we will not need this generalization, as we will be reasoning on a simple one-dimensional example. We will be dealing with pairs of scalars: (*x*, *t*) or (*S*, *F*). Generalization to several space and time variables is not a priori a problem, apart from the cumbersomeness of the formalism. For now, this provides a framework for placing the pairs (*x*, *t*) and (*f*, *g*) on the same footing.

### 2.3. General Conservation Constraints

The systems studied in (finite-time) thermodynamics are subject to finiteness constraints of various kinds. Here, we shall consider constraints on the conservation of matter: together with the equivalent conservation of energy, these are the most fundamental constraints that can be expressed for common physical systems. In the general case, we must write them in the following local form; *f* is a physical quantity defined by its volume concentration, and $\vec{g}$ is a flux vector in three-dimensional space:(1)$$\frac{\partial f}{\partial t}+div\vec{g}=0$$

Knowing that we will restrict ourselves here to laws of the form(2)$$\frac{\partial f}{\partial t}+\frac{\partial g}{\partial x}=0$$
where *f* and *g* are two scalars. We will assume that, due to the granularity of the representation adopted, there is no production term in the mass balances. The preceding relationship can be understood another way: in a relational spirit, the space and time variables *x* and *t*, and more generally the physical quantities *f* and *g*, are not known in themselves but only in their reciprocal relationships: we express the latter by the equality of their coupled variations (partial derivatives) of the type(3)$$\frac{\partial f}{\partial t}=-\frac{\partial g}{\partial x}$$(we keep the—sign to respect the first conservation formulation). Knowing that the dual expression(4)$$\frac{\partial f}{\partial x}=-\frac{\partial g}{\partial t}$$
would also be an admissible law. In the general framework proposed in our work, complete formulations would be of the type$$\frac{\partial f_{x}}{\partial t_{x}}+\frac{\partial f_{y}}{\partial t_{y}}+\frac{\partial f_{z}}{\partial t_{z}}+\frac{\partial g_{x}}{\partial x}+\frac{\partial g_{y}}{\partial y}+\frac{\partial g_{z}}{\partial z}=0$$
which is derived by summation of elementary equalities of the type $\frac{\partial f_{i}}{\partial t_{j}}=-\frac{\partial g_{j}}{\partial x_{i}}$. It can be shown [15,16] that the equations of mechanics and Maxwell’s equations can be put into this form through various changes of variables. See also [17]. The nullity of a transport derivative along an appropriate motion would be yet another way of understanding Equation (2).

For the pair (entropy, entropy flow), we do not have a strict equality in a conservation equation, but the inequality expressing the second law in an open system; the entropy balance is written:(5)$$I=\frac{\partial S}{\partial t}+\frac{\partial F}{\partial x}\;\geq 0$$

And we also write (even if this rarer notation is confusing) the following:$$I=P\left(S\right)=\frac{d_{i}S}{dt}$$
where *d_i_S* refers to the change in entropy due to the phenomena internal (i for internal) to the system (as opposed to the processes in relation with the fluxes).

### 2.4. Lorentz Invariance and Conservation Laws

Lorentz transformation will play hereafter. It allows us to pass from the values (*x*, *t*) (we use one dimension of space and one of time), evaluated in a first frame of reference, to those (*x*’, *t*’) in a second frame of reference moving relative to the first. Subject to various constraints (linearity, isotropy of space, homogeneity of space and time, compliance with the two relativity postulates), we obtain the following results:*x*’ = *ax* + *bt*
*t*’ = *bx* + *at*(6)
with only two coefficients *a* and *b* and not four (expressing a formal symmetry between *t* and *x*; we have taken *c* = 1), which depend on the speed *v* of displacement between the two reference frames, according to(7)$$a=\frac{1}{\sqrt{1-\frac{v^{2}}{c^{2}}}}\;\;\;\;\;\;\;\;\;\;\;b=\frac{-v}{\sqrt{1-\frac{v^{2}}{c^{2}}}}$$

The expressions of the Lorentz transformations in relations (6) and (7) depend on the choice of clocks made. In our new perspective on the theory of relativity, mentioned in the introduction, we highlight the importance of the hidden movements of photons in clocks (which cannot be punctual). Depending on the choices made, various expressions of the Lorentz transformations are possible, as we show in [18], commenting on various results in the literature. Here, we used the standard formulations.

As we shall see, there is a special link between Lorentz invariance and the general form of linear conservation laws, explained above for pairs (*f*, *g*) of two scalars; in Equation (2), we have written:$$\frac{\partial f}{\partial t}+\frac{\partial g}{\partial x}=0$$

Let us now write down the Lorentz transformation for the quantities *f* and *g* linked by law (2). We want this law to be Lorentz invariant, i.e., to be conserved in a change of Galilean reference frame R → R’ (Einstein’s first postulate), with reference frame R’ moving at speed *v* relative to reference frame R. It is shown in [16] that the law(8)$$\frac{\partial f\prime }{\partial t\prime }+\frac{\partial g\prime }{\partial x\prime }=0$$
is also verified, provided we write:*f*’ = *af* + *bg**g*’ = *bf* + *ag*(9)
where the coefficients *a* and *b* describe the Lorentz transformations for *x* and *t*. Equations (2), (6) and (7) have been used (the Lorentz transformation assumes the second postulate of relativity). We, therefore, obtain transformation laws for physical quantities that are identical to the transformations for space and time coordinates (these results can be found in the developments of standard relativity, even if it is a little hidden; we give this result a general value. Expressed in the standard covariant notation adopted in relativity, any conservation law can be expressed as: ∂_i_J_i_ = 0, where J is the suitable 4-vector. When expressing it in this form, it is immediately clear that a Lorentz transformation would not change the form of the law).

The reciprocal holds. If the transformation laws of type (6), (7) and (9) are verified, then the laws (2) linking *f* to *g* and (8) linking *f*’ to *g*’ are such that(10)$$\frac{\partial f}{\partial t}+\frac{\partial g}{\partial x}=\frac{\partial f\prime }{\partial t\prime }+\frac{\partial g\prime }{\partial x\prime }$$

This last result expresses both the form of the law sought and its invariance under the action of the change of reference frame; if we note that relations (10) are valid for all forms of functions *f* and *g*, they are particularly verified for functions *f* and *g* equal to zero, and the common value of the two expressions in relation (10) is zero. This is written as:(11)$$\frac{\partial f}{\partial t}+\frac{\partial g}{\partial x}=\frac{\partial f\prime }{\partial t\prime }+\frac{\partial g\prime }{\partial x\prime }=0$$

Thus, *a necessary and sufficient condition for a law to be Lorentz invariant *(the Lorentz transformation being given for *x* and *t*) *is that it has the specified form of type (2)* or a mathematically equivalent form (at orders derived or integrated with respect to the original formulation). The characteristic of this form is to equalize derivatives of quantities with respect to time with derivatives of coupled quantities with respect to space. This is the case for what we call the basic laws of physics. As mentioned above (note (3)), this is the case for Maxwell’s equations and the equations of mechanics usually tested for Lorentz invariance. If the form of the equations confers Lorentz invariance, as is well known, we emphasize here the reciprocal proposal and its generality. This result expresses *a strong link between relational thinking (we spoke of relational epistemology) and Lorentz invariance.* It also shows that the pair (*x*, *t*) is on the same footing as pairs of physical quantities in duality and behaves in the same way or, conversely, that pairs of type (f, g) behave like pairs (x, t). This is a way of saying that space and time can be defined by the measurements of f and g. This will allow us to understand the expressions “finite time” and “finite resources” as having a real mathematical equivalence (and not just a practical one), which is one of the objectives of our study (see Section 3.5).

### 2.5. Entropy Balance

The same applies to entropy (by formally replacing *f* and *g* by *S* and *F* in the linear conservation relation)(12)$$\frac{\partial S}{\partial t}+\frac{\partial F}{\partial x}=\frac{\partial S\prime }{\partial t\prime }+\frac{\partial F\prime }{\partial x\prime }$$In particular, if(13)$$\frac{\partial S}{\partial t}+\frac{\partial F}{\partial x}\;\geq 0$$We will have(14)$$\frac{\partial S\prime }{\partial t\prime }+\frac{\partial F\prime }{\partial x\prime }\;\geq 0$$With again*S*’ = *aS* + *bF*F’ = *bS* + *aF*
(15)
where *a* and *b* are the coefficients of the Lorentz transformations given by (7). In the relativistic thermodynamics literature, it is generally accepted that the entropy scalar function *S* is a relativistic invariant, i.e., *S*’ = *S*, although it is pointed out that consensus on the issue does not seem to have been reached (see in the numerous references [19,20,21,22,23,24,25,26,27,28,29,30,31,32,33,34,35,36,37,38,39,40,41,42,43,44] along with general considerations on relativistic thermodynamics).

For the general reasons we have given we think it useful to embed *S* in four vectors (*S*, *F*). In short, we mustn’t separate physical quantities from the laws in which they are involved, in this case laws involving the simultaneous time and space derivatives of dualistic quantities. It’s a way of joining up with the requirements of quantum mechanics concerning the absence of rigorously isolated systems. We are then led to(16)$$P\left(S\right)=\frac{\partial S}{\partial t}+\frac{\partial F}{\partial x}=P\left(S^{\prime }\right)=\frac{\partial S\prime }{\partial t\prime }+\frac{\partial F\prime }{\partial x\prime }\geq 0$$
and relativistic invariance concerns entropy production, not entropy alone. The optimization criterion we are interested in here (see below) is that given by (13), and, in the study of reference frame changes, we will rely on relation (16).

## 3. Results

### 3.1. Study of an Example

Let us turn now to the concrete example already suggested. We assume that the balance of a physical quantity *f* is governed by the following conservation equation, analogous to (2) seen above:$$\frac{\partial f}{\partial t\;}+\frac{\partial g}{\partial x\;}=0$$
where *g* represents the flux of the quantity *f* (recall that the problem is posed in one dimension of space *x* and one of time *t*). This equation simulates the behavior of a rock defined by the concentration *f* (related to the unit length) of a single chemical constituent C (e.g., iron) in the solid phase ([45,46]; ion-exchange columns provide an analogous problem [47]). The rock is traversed in its porosity *p* (assumed to be low) by an *aqueous fluid* in disequilibrium, characterized by the concentration *g* in the fluid. The flow involves the velocity *w* of the percolating fluid and the porosity *p* in the complete expression *pwg*. We will assume that we have normalized the quantities so that *pw* (Darcy velocity) = 1, and a velocity is, thus, hidden in *g*. Then, the velocity w of the fluid is assumed to be uniform, constant, and non-dimensional. This somewhat surprising choice is made for practical reasons, in order to respect the simple form of the continuity equation of type (2).

Another surprising choice on our part is the inclusion of our example in a relativistic thermodynamics framework. This physical example is not likely to involve anything moving at relativistic speeds. Without fast particles, it is not immediately clear how “relativistic thermodynamics”, the theoretical structure used here, could be tested. Would ordinary Newtonian/Galilean mechanics, perhaps in its diffusion form, not suffice? In response to this natural objection, we would say that we are using the word relativistic in a broad sense, certainly in continuity with and consistent with the standard meaning: we speak of relativity to refer to the general structure of our understanding of space and time, extending beyond the realm of physics and into the human and social sciences. It points the way to a close relationship between time and space. Relativity allows us to discuss the behavior of space and time, which manifest themselves in interesting ways when we consider relative movements between reference frames (refer to our introductory remarks).

We will also assume that a local equilibrium is achieved between the solid and the fluid, according to a law, called an isotherm, not necessarily linear, *g* = *g*(*f*). By normalizing the quantities appropriately, we assume that *g*(0) = 0 and *g*(1) = 1 (Figure 2).

We impose a disequilibrium on the system with the following contrasted initial and boundary conditions (Figure 3): the fluid arrives at *x* = 0 with *g* = 1 in a rock characterized by *g* = 0 for t = 0 from abscissa 0 to infinity to the right. We are considering a reactor of size *L*. We are interested in the finite time *T* for complete transformation of the reactor to the value 1 of the concentrations. This will correspond to a finite content *Φ* of the component inside the reactor and an integrated flux necessary for transformation *Γ*.

Various scenarios are possible, described by *g*(*x*, *t*) distributions. For reasons of generality and because it is interesting and fruitful to do so (see introduction), we pose the problem in terms of distributions in the mathematical sense (weak solutions). Multiple, discontinuous solutions (also called shocks) may indeed be encountered; continuous solutions are called rarefaction waves or *détentes*. We need to impose an optimization criterion that selects the weak solutions that make physical sense. Let us insist: it is an important property of hyperbolic problems when looking for weak solutions in the mathematical sense, i.e., representing true discontinuities in the sense of distribution theory, to present multiple solutions (non-unique). Solutions are complete spatio-temporal evolutions *g*(*x*, *t*) linking boundary conditions as specified (Riemann problem). To do this, we postulate the existence of a (entropy, entropy flux) pair, i.e., (*S*, *F*), where *S* depends on the quantity *g* (or, what is equivalent, to *f*, thanks to local equilibrium), verifying (see the theory of hyperbolic problems [48,49]):$$P\left(S\right)=\frac{\partial S}{\partial t}+\frac{\partial F}{\partial x}\;\geq 0$$

There are weak formulations of the previous entropy condition. It also corresponds to an extremum [50]. This is in line with [3]. The local equilibrium condition can be written as [45]:$$\frac{\partial S}{\partial f}=\frac{\partial F}{\partial g}$$

Hyperbolic problem theory teaches that the velocities of progression of different concentrations in space are proportional to the slopes of the isotherm at the corresponding points (scalar case). Similarly, shocks or discontinuities between various points have velocities proportional to the slopes connecting the points concerned (cf. [48,51]). Refer to the caption in Figure 2.

### 3.2. Solving the Problem

Figure 4, Figure 5 and Figure 6 show various weak solutions to the problem posed in Figure 3. In Figure 4, two compositional shocks or discontinuities propagate through the system. The shock leading from the starting concentration (*f* = *g* = 0) to that of point A moves more rapidly than the shock seeing the concentration change from that of point A to that of point B. The concentration at point A can have any value between 0 and 1. In Figure 5, after an initial shock similar to that in the previous figure, we see a rarefaction wave between A and B. Figure 6 shows the propagation of a single shock going directly from level 0 to level 1. The various profiles satisfy the shock conditions just described.

Figure 7 shows a rarefaction wave between point O (0, 0) and point B (1, 1). Only this rarefaction wave meets the entropy condition. In fact, this profile can be understood as meeting a stability constraint: if a fluctuation causes an intermediate composition between 0 and 1 to appear, this composition is maintained and propagated in the sequence of compositions, because its speed is between the speeds of the upstream and downstream compositions (the profiles in the previous figures are not stable, as they cannot avoid the propagation of intermediate compositions between the compositions of the initial condition and the composition of the boundary condition, should these appear through fluctuation). More generally, practitioners of hyperbolic problems trace paths in the isotherms connecting the extreme points of the boundary conditions. In the general case, and for isotherms with more complicated shapes than that in this study, the optimal path in the sense of the entropy condition is given by the convex envelope to the isotherm between the extreme points.

### 3.3. “Finite” Quantities

On the basis of the above results, we can define various quantities characterizing the finiteness of the problem: time *T* for the complete transformation of the reactor of size *L*. We can see from the above that this is the time it takes for the composition B to pass through the reactor from abscissa 0 to abscissa *L*. Depending on the case, the speed of B is not the same, particularly if B is produced after a shock (non-physical solution, Figure 4 and Figure 6) or at the end of a rarefaction wave (physical solution, Figure 7, or partially physical, Figure 5).

The times *T* are as follows, for Figure 4, Figure 5, Figure 6 and Figure 7, respectively (we call *T*_4_ the transformation time corresponding to the propagation shown in Figure 4, and similarly for other times):

*T*_4_ = *L*/*v_AB_* where *v_AB_* is the velocity of the shock passing from A to B

*T*_5_ = *L*/*v_B_* where *v_B_* is the velocity of point B (tangent to the isotherm)

*T*_6_ = *L*/*v_OB_* = *L* where *v_OB_* = 1

*T*_7_ = *L*/*v_B_* = *T*_5_

We can also calculate the total quantity *Φ* of chemical component C contained in the reactor of dimension *L* and the finite flux *Γ* that was required for the transformation. Here, we see that *Φ* = *Γ* = *L*. This particular result originates from the initial condition where *g*(*t* = 0) = 0.

### 3.4. Moving Frame: Transformation of Equations

Let us now consider a moving frame of reference R’, relative to the frame of reference R used to solve the problem described in the previous section. Let *v* be the relative velocity between the two reference frames.

In our example, there seems to be a clear preferred frame of reference in the form of the rock medium in which the water carrying the atoms of the chemical element we are interested in (iron, for example) moves. This is clearly a specific physical frame of reference, and with this frame of reference present, it is simply not clear how Lorentz invariance could be appropriate. There are two answers to this question. The first is that the reference frames used in standard relativity all have distinctive features that make each one privileged in its own way (in the example of Langevin’s twins, the reference frame at rest relative to the Earth, the earth itself, is quite special). Second, we can say that we are operating at a mathematical level and that the two reference frames R and R’ have no other characteristic at this level than that of having a relative displacement. From a general point of view, in the geological example, we can easily imagine other reference frames in relative motion (such as the fluid in migration).

We will bring into play the relativistic approach described in the first section and ask ourselves a series of questions:-What will be the new conservation equation linking the physical quantities in the moving frame of reference?-What will be the new optimization criterion?-What relationships can we write between the quantities *x*, *t*, *f*, *g*, *S*, *F* (frame R) and the quantities *x*’, *t*’, *f*’, *g*’, *S*’ and *F*’ (frame R’)?-How will the finite quantities *L* and *T*, as well as *Φ* and *Γ*, which relate to the (finite-time) thermodynamics problem, be transformed?-Can we write relationships between these different finite quantities?

These questions can be answered simply for this particular example, thanks to the results of the first part. The conservation Equation (2) is Lorentz invariant and the new equation is identical to (8) above$$\frac{\partial f\prime }{\partial t\prime \;}+\frac{\partial g\prime }{\partial x\prime \;}=0$$

And, as far as the optimization principle is concerned, it is also Lorentz invariant and we will have the condition given by (16)$$P\left(S\prime \right)=\frac{\partial S\prime }{\partial t\prime }+\frac{\partial F\prime }{\partial x\prime }\;\geq 0$$

We are, therefore, in the simple situation of having the same Lorentz transform formulas for both space and time variables and for concentrations, as in (6) and (9).*x*’ = *ax* + *bt*
*t*’ = *bx* + *at*
and*f*’ = *af* + *bg**g*’ = *bf* + *ag*
where *a* and *b* are the coefficients given by relation (7). We will not dwell on what concerns the primary physical quantities involved in the Lorentz transformations just written. In our research on (finite-time) thermodynamics, we will take a closer look at what concerns the finite quantities *L*, *T*, *Φ* and *Γ*.

### 3.5. Moving Reference Frame: Transformation of Finite Quantities

The transformation of space amplitudes *L* is given by the following classical formula:*L* = *L*’/*a*
(17)
with$$a=\frac{1}{\sqrt{1-\frac{v^{2}}{c^{2}}}}$$where *a* is greater than 1. This relationship expresses that a length *L*’ evaluated in the moving frame of reference is seen contracted from the fixed frame of reference. For durations *T*, we also have, according to the classic formula*T* = *aT*’(18)
where, this time, the durations evaluated *T*’ in the moving frame of reference are seen dilated from the fixed frame of reference. From the above relationships, we derive*LT* = *L*’*T*
(19)

or(20)$$\frac{L}{L\prime }\frac{T}{T\prime }=1$$
which expresses that the relative variations of lengths and durations are correlated. This relationship has been commented on in the literature. In the case of three dimensions of space, rather than “conservation of a hypervolume” [52], we prefer to see it as that of a norm Σ*t_i_x_i_*, see [53].

Let us now look at how the quantities *Φ* and *Γ* are transformed. To do this, let us return to the first postulate of relativity: the laws of physics are the same in the two reference frames in Galilean motion relative to each other. Just as quantities *Φ* are evaluated in proportions of lengths *L*, quantities *Φ*’ will be evaluated in proportions of lengths *L*’, in accordance with the following ratios*Φ*/*L* = *Φ*’/*L*’
(21)


For *Γ* flows, we can similarly say that they are evaluated by *T* durations and that the following relationships must be respected*Γ*/*T* = *Γ*’/*T*’
(22)


Relationships (21) and (22) could just as well be understood by saying that *Φ* and *Γ* provide the units of space and time. By multiplying the two previous relationships, we obtain*Φ Γ*/*LT* = *Φ*’ *Γ*’/*L*’*T*’
(23)


By virtue of *LT* = *L*’*T*’ (19), this gives*Φ Γ* = *Φ*’ *Γ*’
(24)


This relationship expresses that the relative variations of the finite quantities *Φ* and *Γ* are also correlated. This expression could be obtained independently if we decide to measure space and time using the quantities f and g. In both cases (by direct independent determination or indirectly via Equations (19), (21), and (22)), the expression *Φ Γ* = *Φ*’ *Γ*’ remains the same, even if the specific values of *Φ*, *Γ*, *Φ*’, *Γ*’ that verify it differ.

Other similar relationships can be obtained. Carrying (17) into (21), we obtain*Φ* = *Φ*’/*a*
(25)


Using (18), this gives*ΦT* = *Φ*’*T*’
(26)

which we can also write(27)$$\frac{\Phi }{\Phi \prime }\frac{T}{T\prime }=1$$

Analogous reasoning yields the relationship*ΓL* = *Γ*’*L*’
(28)

which we can also write(29)$$\frac{\Gamma }{\Gamma \prime }\frac{L}{L\prime }=1$$

This expresses a complementarity between finite resources (reactor contents, summation of the flux supplied) and finite space and time. The size of the systems mobilized and the possible durations of the reactions concerning them have something to do with the quantities of resources involved, directly or contributed. We believe that the results expressed in relations (26) to (29) are important, even from a purely qualitative point of view. They place (finite) time and space on the same level as (finite) material flows and resources. They were obtained thanks to the theory of relativity, but this theory could now be forgotten. However, it has fully played its role, which we see as fundamentally that of linking the construction of space and time variables to the comparison of concrete material processes at work in physical reality.

## 4. Discussion and Conclusions

At the end of this work, let us sketch out a few conclusions: the future will tell us what general value they may have, having been obtained by studying an example. We can use them as a basis for further fundamental research, to be carried out over the long term. In summary, the relationship between relativity and thermodynamics in this work is based on the fact that the thermodynamic quantities *f* and *g* are treated as relativistic, even though the velocities are low. It is also based on the combination of relationships involving the pairs (*f*, *g*) and (*x*, *t*), which highlight the organic relationships between finite resources and finite time. The relationship between finite-time thermodynamics and hyperbolic problems is based on the fact that there are multiple weak solutions and that a thermodynamic criterion is used to select them.

The constraints brought by the relativistic functioning as influencing the thermodynamic functioning, particularly in the use of four vectors, have oriented us, as far as the optimization criterion is concerned, towards the entropy production. Entropy alone is not a relativistic invariant.

We have seen the complementary-type relationships between finite resources and finite time and space. Finitudes are linked between space and time, on the one hand, and resources and their flows, on the other. *If time is finite, so are resources, which can be neither zero nor infinite.* This is a way of reiterating that space and time cannot be thought of on their own and are a way of understanding the phenomena associated with material resources. This is a way to explore dimension 2 in Figure 1. Along these lines, we can imagine posing the problem differently, by exchanging the roles of the variables (*x*, *t*), on the one hand, and (*f*, *g*), on the other, in the mathematical equations, in particular the conservation Equation (2) from which we start. The equivalence between the two approaches expresses the equivalence between time/space and flows/quantities of matter.

With regard to the role of hyperbolicity (dimension 1 in Figure 1) for large systems, we have seen that neglecting diffusion allows us to obtain finite propagation speeds and ensure a finite time that depends directly on them. The trade-off for this choice is the multiplicity of weak solutions for *g*(*x*, *t*) evolutions.

We mentioned two research directions, labeled 3 and 4, in Figure 1. To proceed in direction 3, we must resume the hyperbolic approach by exchanging the variables (*x*, *t*) with the variables (*f*, *g*). These mark space and time, and the old scale of space and time is distorted. To proceed in direction 4, we must introduce weak solutions into the Lorentz transformation. A line of research into these questions is provided by the works of Andresen and Essex ([54] and other references therein). These authors are interested in what we are able to see of systems, depending on the comparative speeds of the processes taking place within them and the time we spend observing them. Thus, we can say that when scanning a spatial system, the abrupt variations found within it may or may not be considered as discontinuities depending on the speed of the scan (i.e., the temporal pixel). If the scan is slow or defined with a sufficient temporal resolution, we will see the intermediate states of the abrupt variations and will not consider them as real mathematical discontinuities. At faster observation speeds, we will not see the intermediate states (and with larger temporal pixels): we will then conclude that discontinuities are present. If we now compare the views we have of a system from two reference frames in relative motion, we conclude from relativistic effects relating to time dilation that the question of discontinuities will not arise in the same way in the two frames (the question may arise in one frame but not in another). All this must be clarified in appropriate research.

We have also seen that relativistic changes in conservation constraints and optimization criteria are analogous. Can we see these two types of mathematical relations as the expression of one and the same point of view? We can think of Kruzhkov’s [55] formulation of hyperbolic problems, where conservation and entropy conditions derive from a single mathematical writing. We may also think of the formulations of the two principles of thermodynamics as forms of a single principle entitled “stable-equilibrium-state-principle” according to [56] (see also [57]) from which, out of equilibrium, formulations are derived.

Finally, the results we obtained give us food for thought on the relationships between thermodynamics and relativity. We can indeed write complementarity-type relations linking entropy variations and its flux to spatial and temporal quantities (identical to (26) and (27)). Is this a way of understanding the necessity for homogeneity, of both spatio-temporal origin (based on regularly spaced graduations, in the postulate c = cte) and thermodynamic origin in the second law? In [58], we laid propositions in this direction.

## Figures and Tables

**Figure 1 entropy-27-00700-f001:**
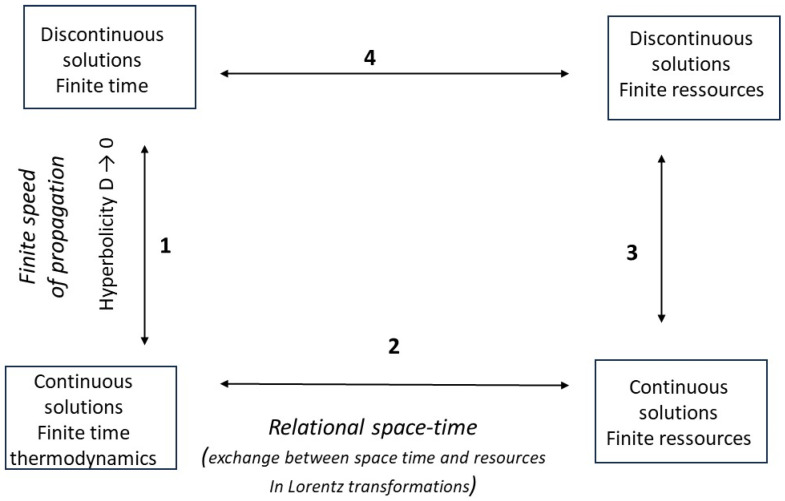
Logical structure of our research.

**Figure 2 entropy-27-00700-f002:**
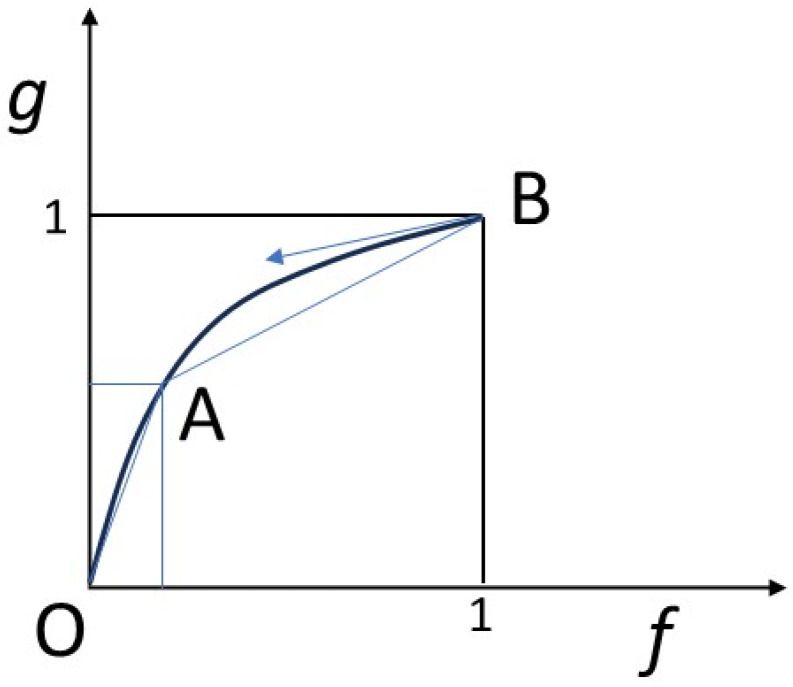
Isotherm linking rock and fluid concentrations at equilibrium. The concentration *f* of the chemical constituent in the solid is shown on the abscissa, that in the fluid, *g*, on the ordinate. Concentrations are normalized so that the amplitude of variation is between 0 and 1 in the solid and in the fluid respectively. The isotherm is concave downwards and connects points O (0, 0) and B (1, 1). An intermediate point A is shown. Hyperbolic problem theory teaches that the velocities of progression of different concentrations in space are proportional to the slopes of the isotherm at the corresponding points. So, the arrow drawn from point B, tangent to the isotherm at this point, gives the speed of progression of concentration (1, 1). Similarly, shocks or discontinuities between various points have velocities proportional to the slopes connecting the points concerned [48]. So, a shock between point (0, 0) and point A will have a velocity along OA, and the shock between point B and point (1, 1) will have a velocity along AB.

**Figure 3 entropy-27-00700-f003:**
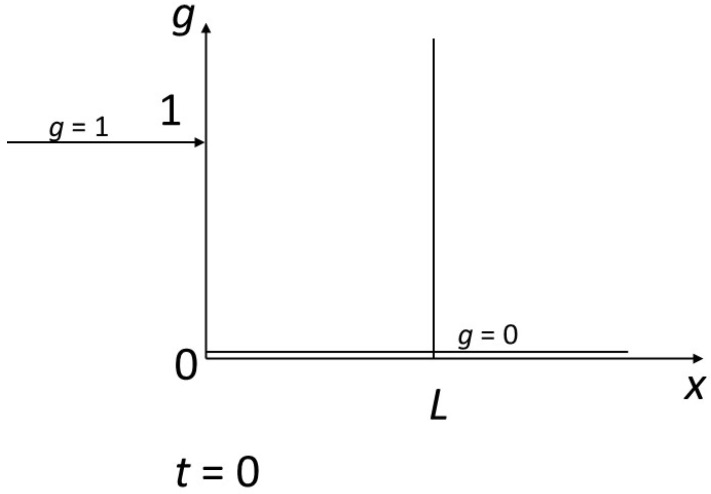
Representation of the reactor studied. The reactor of finite length *L* is represented along the *x*-axis. The fluid enters from its left side at *x* = 0, and leaves to the right from *x* = *L*. The initial concentration in the fluid contained in the reactor (and along the *x*-axis) is *g* = 0, the incoming concentration at *x* = 0 is *g* = 1. The corresponding equilibrium concentrations for the solid are *f* = 0 and *f* = 1, respectively.

**Figure 4 entropy-27-00700-f004:**
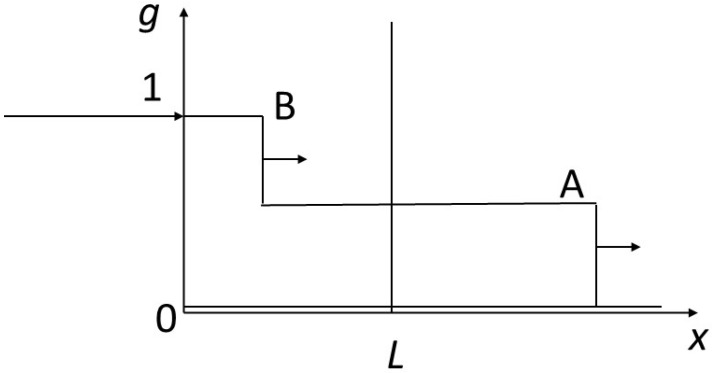
Representation of the reactor crossed by two composition discontinuities. The state of the reactor at a given moment has been represented, traversed by a composition discontinuity leading from *g* = 0 to point A, followed by another discontinuity connecting point A to point B; points A and B are shown in Figure 2. Applying the rules governing velocities recalled in the caption of Figure 2, we can see that the velocity of the first front is almost six times faster than that of the second. The arrow perpendicular to the vertical discontinuities indicates the direction of the corresponding fronts (to the right).

**Figure 5 entropy-27-00700-f005:**
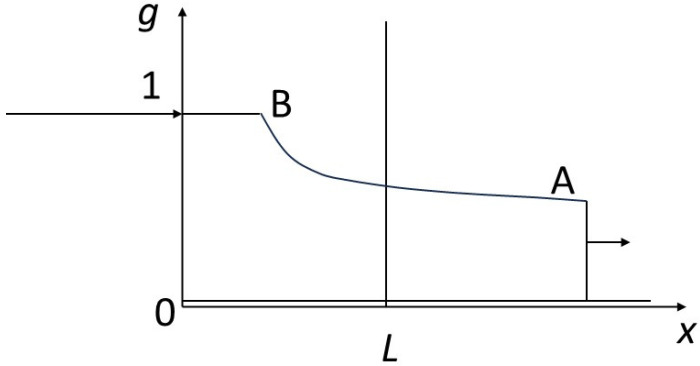
Representation of the reactor pervaded by a shock and a rarefaction wave. By comparison with the previous figure, the discontinuity between 0 and A is kept, but there is a rarefaction wave between A and B. The advance velocity of B is given by the tangent to the isotherm at the corresponding point (see Figure 2). The arrow perpendicular to the vertical discontinuity indicates the direction of the corresponding front (to the right).

**Figure 6 entropy-27-00700-f006:**
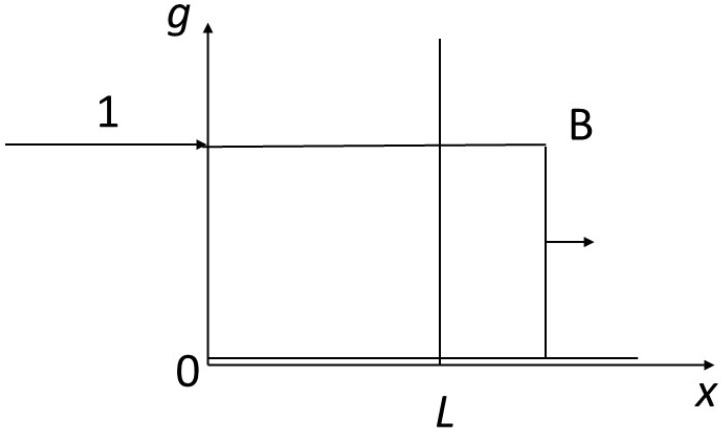
Representation of a reactor traversed by a single shock. Here, we have the case of a single discontinuity between point O (0, 0) and point B (1, 1) with a velocity equal to 1, i.e., the slope of the segment connecting the two points on the isotherm (Figure 2). The arrow perpendicular to the vertical discontinuity indicates the direction of the corresponding front (to the right).

**Figure 7 entropy-27-00700-f007:**
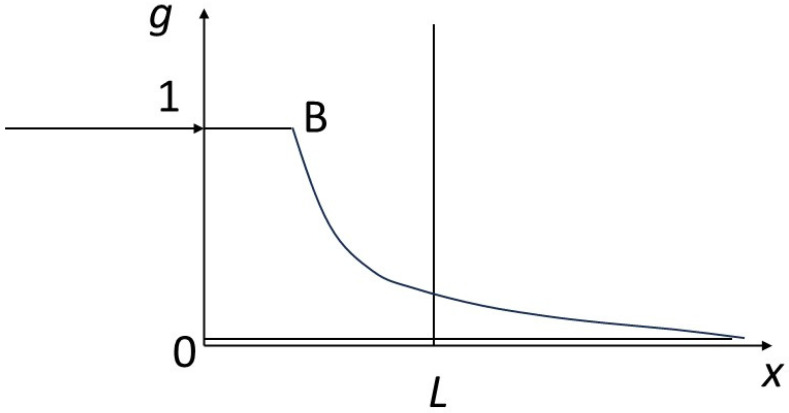
Representation of the reactor through which a composition rarefaction wave takes place. In contrast to Figure 4, Figure 5 and Figure 6, the reactor here is traversed by a complete rarefaction wave linking point (0, 0) and point (1, 1) of the isotherm (Figure 2). The first velocities (for the smallest concentrations), visible on the slopes of the isotherm, are faster than those for the largest concentrations, which arrive behind (lower slopes) and are therefore ahead, generating the rarefaction wave. Only this configuration respects the entropy condition and makes physical sense.

## Data Availability

There are no data.

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
