# Peer review of "(Finite-Time) Thermodynamics, Hyperbolicity, Lorentz Invariance: Study of an Example"

_entropy, 2025, doi:10.3390/e27070700_

Round 1

Reviewer 1 Report

Comments and Suggestions for Authors

Referee report of “Finite time thermodynamics and Lorentz invariance. Study of an example,” by Bernard Guy, MS entropy-3623795. Summitted to the journal Entropy.

In my opinion, this manuscript suffers from several problems that make it difficult for me to recommend it for publication in Entropy.

First, I am not sure how to characterize this manuscript. Is it a research article, a review, or a teaching article? Relativistic thermodynamics, where thermodynamic variables and their fluxes are combined into four vectors, is hardly a new subject, so the extended introduction of this topic seems better placed in a teaching journal. I don’t think that the research-oriented journal Entropy qualifies. But with only 26 references, this manuscript hardly qualifies as a review either. Likewise, no original research seems to be presented here, so this does not appear to be a research article. Readers always want to know what type of paper they are reading.

Second, in the third section of the manuscript, the author presents their main result, a “simple example based on geology.” They consider a rock containing a single constituent present in some concentration. The example given of the constituent is iron. The rock is considered to be a solid, with low porosity to the iron, which is regarded to be a fluid. But I would consider this physical situation to present as a diffusion process, and not one of fluid flow through some medium. And, in either case, the transfer velocity would be very low, far from fast enough for relativity to be even remotely relevant. And if the velocity were relevant, doesn’t the rest frame of the rock constitute a special reference frame, one breaking the symmetry among the reference frames always assumed in the Lorentz transformations?

In my opinion, the issues that I have raised here weigh seriously against any recommendation to publish in Entropy.

Author Response

Reviewer number 1

In my opinion, this manuscript suffers from several problems that make it difficult for me to recommend it for publication in Entropy.

First, I am not sure how to characterize this manuscript. Is it a research article, a review, or a teaching article? Relativistic thermodynamics, where thermodynamic variables and their fluxes are combined into four vectors, is hardly a new subject, so the extended introduction of this topic seems better placed in a teaching journal. I don’t think that the research-oriented journal Entropy qualifies. But with only 26 references, this manuscript hardly qualifies as a review either. Likewise, no original research seems to be presented here, so this does not appear to be a research article. Readers always want to know what type of paper they are reading.

Answer:

I thank reviewer number one for his/her work.

This paper is a research article. This must be more clearly stated. I am adding a new part in the introduction. “It seems hardly a new subject”: yes, but there are still some controversies. I discuss this in the new part in the introduction, and I am adding new references about relativistic thermodynamics.

“No original research”? May I understand these words as: “nothing is wrong” or “nothing is false” in the paper? Is’nt it already comforting for me ? I do not pretend to know all the bibliographic references, but, I would be interested to know whether relations such as equations (26) or (28) in my paper may be found elsewhere?

Second, in the third section of the manuscript, the author presents their main result, a “simple example based on geology.” They consider a rock containing a single constituent present in some concentration. The example given of the constituent is iron. The rock is considered to be a solid, with low porosity to the iron, which is regarded to be a fluid. But I would consider this physical situation to present as a diffusion process, and not one of fluid flow through some medium. And, in either case, the transfer velocity would be very low, far from fast enough for relativity to be even remotely relevant. And if the velocity were relevant, doesn’t the rest frame of the rock constitute a special reference frame, one breaking the symmetry among the reference frames always assumed in the Lorentz transformations?

In my opinion, the issues that I have raised here weigh seriously against any recommendation to publish in Entropy.

Answer

I give further information about the example in the introduction. The chemical component (e.g. iron) is dissolved in the permeating water. The important in the example chosen is not that it comes from geology, but that is a hyperbolic problem (see ref. [44] in the updated list of references); these problems are not enough known among the thermodynamics community. The notion of scale is fundamental: for large systems, diffusion becomes negligible with respect to advection. The mathematical structure is important, and this is not a question of velocity of light in relativity theory; I discuss this in the new part in the introduction (also answering to comments from other reviewers). As for the meaning of the rest frame, I won't dwell on this point: in abstract terms, a frame is simply a way of designating pairs (x, t).

Reviewer 2 Report

Comments and Suggestions for Authors

The paper considers the interesting problem of formulating Lorentz invariant version of the finite-time thermodynamics. The relativistic thermodynamics was once a subject of great discussion and controversy (for example, [1, 2]).

The conclusion made in the earlier works on the subject (as early as in the seminal book by Tolman [3]) is that the main problems of relativistic thermodynamics is of philosophical not physical kind.

So I was excited when I first saw the title of the manuscript under review, because, as I think, some philosophical questions of relativistic thermodynamics become easier when one works in the finite-time framework. However, the author failed to satisfy my expectations of this kind: there is no clear formulation of the problem in the manuscript and there is no strong philosophical or logical foundations. Even the notion of two thermodynamics systems being in the relative motion is highly controversial and the author did not even mention this. Also there is not a single mention of the temperature in the manuscript, while the temperature was the most debated quantity with respect to the Lorentz invariance [1]. I insist that one can not formulate the relativistic version of FTT without even taking the temperature into the account.

The example that the author provides is taken from the geology and it is not clear, how it is related to the relativistic thermodynamics. In my opinion, the author provided some reduced version of the relativistic continuum mechanics in the spirit of the book by Tolman [3], mentioned above, but there is no content in the manuscript that is directly related to the FTT.

I hope, however, that the manuscript could be improved and published in the Special Issue. To do this the author should consider to:

  1. Properly identify all philosophical and logical foundations of the theory he tries to develop.
  2. Develop the Lorentz invariant version of the finite-time or irreversible thermodynamics.
  3. Show that the finite-time formulation of the relativistic thermodynamics is less controversial than the classical one.
  4. Provide the example in the spirit of the finite-time thermodynamics
  5. Give the proper references. At this time the manuscript does not include references to some classic works in the field.

I see this as a major revision of the paper. After the mentioned revision the paper could be published in the Special Issue of the Entropy.

  1. H. Callen, G. Horwitz, “Relativistic thermodynamics”, UFN, 107:3 (1972), 489–502; Am. J. Phys., 39 (1971), 938–947
  2. N.G. van Kampen, “Relativistic Thermodynamics of Moving Systems”, Phys. Rev., 173:1, 295–301 (1968).
  3. R. Tolman, “Relativity, Thermodynamics and Cosmology”, Oxford, Clarendon Press, 1934.

Author Response

Reviewer number 2

The paper considers the interesting problem of formulating Lorentz invariant version of the finite-time thermodynamics. The relativistic thermodynamics was once a subject of great discussion and controversy (for example, [1, 2]).

The conclusion made in the earlier works on the subject (as early as in the seminal book by Tolman [3]) is that the main problems of relativistic thermodynamics is of philosophical not physical kind.

So I was excited when I first saw the title of the manuscript under review, because, as I think, some philosophical questions of relativistic thermodynamics become easier when one works in the finite-time framework. However, the author failed to satisfy my expectations of this kind: there is no clear formulation of the problem in the manuscript and there is no strong philosophical or logical foundations. Even the notion of two thermodynamics systems being in the relative motion is highly controversial and the author did not even mention this. Also there is not a single mention of the temperature in the manuscript, while the temperature was the most debated quantity with respect to the Lorentz invariance [1]. I insist that one cannot formulate the relativistic version of FTT without even taking the temperature into the account.

Answer

I thank reviewer number 2 for his/her work.

I am adding new references, including that proposed by the reviewer. I am writing a new part in the introduction. I explain there how I understand, from a very general (philosophical) standpoint, thermodynamics, finite time thermodynamics, relativity, and the articulation with the example chosen: this example is important for its “hyperbolicity” which is one important ingredient to the paper. This point must be stressed as it has not been seen by the reviewers. The formulation of the problem is then better explained. The status of temperature is also briefly commented.

The example that the author provides is taken from the geology and it is not clear, how it is related to the relativistic thermodynamics. In my opinion, the author provided some reduced version of the relativistic continuum mechanics in the spirit of the book by Tolman [3], mentioned above, but there is no content in the manuscript that is directly related to the FTT.

Answer

The new part in the introduction tries to answer these comments. The important in the example chosen is not that it comes from geology, but that is a hyperbolic problem; these problems are not enough known among the thermodynamics community. The notion of scale is fundamental: for large systems, diffusion becomes negligible with respect to advection. The mathematical structure is important, and this is not a question of velocity of light in relativity theory; I discuss this in the new part in the introduction (also answering to comments from other reviewers).

I hope, however, that the manuscript could be improved and published in the Special Issue. To do this the author should consider to:

  1. Properly identify all philosophical and logical foundations of the theory he tries to develop.

Answer: see the new lines in the introduction?

  1. Develop the Lorentz invariant version of the finite-time or irreversible thermodynamics.
  2. Show that the finite-time formulation of the relativistic thermodynamics is less controversial than the classical one.
  3. Provide the example in the spirit of the finite-time thermodynamics

Answer: I am afraid I do not fully understand; this means that I don't know enough about finite-time thermodynamics?

  1. Give the proper references. At this time the manuscript does not include references to some classic works in the field.

I see this as a major revision of the paper. After the mentioned revision the paper could be published in the Special Issue of the Entropy.

  1. Callen, G. Horwitz, “Relativistic thermodynamics”, UFN107:3 (1972), 489–502; Am. J. Phys.39 (1971), 938–947
  1. N.G. van Kampen, “Relativistic Thermodynamics of Moving Systems”, Phys. Rev., 173:1, 295–301 (1968).
  2. R. Tolman, “Relativity, Thermodynamics and Cosmology”, Oxford, Clarendon Press, 1934.

Answer

I have added a great deal of new references, there were not enough.

Reviewer 3 Report

Comments and Suggestions for Authors

Please find my comments in the attached file.

Author Response

Reviewer number 3

1

The paper presents a case study of a finite-time-thermodynamic system governed by a first

order partial differential equation in time t and one spatial dimension x. The system

described is a percolating fluid characterized by variable concentration of a given

substance. The main questions addressed by the work concern the correct treatment of

this system under a Lorentz transformation and the connection with the laws of

thermodynamics.

The work is quite interesting and well written. However, there are some aspects that could

be improved, in my opinion.

A general comment: when using variables’ symbols in a line of text, could you please use

italic type? E.g. at line 204: “velocity w of the percolating fluid and the porosity p in the

complete expression p w g”. And the same comment applies to all the equations.

Answer: I thank reviewer number 3 for his/her work

All the variables have been put in italic type throughout the whole text and the figures.

Line 37: you mention here for the first time that the example considered is from geology.

At first glance, this seems to be quite surprising, as most geological processes happen on

time-scales that are very far from the relativistic regime. Maybe you could add a short

comment about that.

Answer: a comment is given in the new part in the introduction

Line 74, sentence starting with “To space, …”. I don’t understand this sentence. It sounds

like the last part of the sentence has been forgotten. Or perhaps the word “that” should be

removed?

Answer: the sentence is corrected

Line 80, sentence starting with “From…” the formula there does not look familiar to me. Is

this another way of expressing invariance of the relativistic distance? Could you clarify?

Answer: no, this is a way of constructing the usual time as a scalar. The relativistic distance could also be written with the three intermediate pre-time coordinates. I am appending a footnote on the subject.

Line 118: I am not sure I understand the equation. Is the right-hand side ∂_i S^i = 0, where

S is the four-vector (S,F_x,F_y,F_z). In other words, is this just a compact way of expressing

the continuity equation for S? Could you please clarify your notation or perhaps use a

more standard notation?

Answer: I am adding a sentence explaining the notation

Line 123: you write “constraints that we won't repeat here”. Could you please provide a

reference that you would advise a reader to consult?

Answer: I am adding a list of these constraints

Lines 146 and 147: I believe it would help many readers if these results were also

expressed in the standard covariant notation adopted in relativity. Any conservation law

(e.g. charge conservation) can be expressed as: ∂_i J^i = 0, where J is the suitable 4-vector.

When expressing it in this form, it is immediately clear that a Lorentz transformation Λ

would not change the form of the law.

Answer: I am adding a footnote on the subject

Line 150: The right-hand side of this equation is identical to the left-hand side. In other

words, the equation as written now is just a trivial identity. Could you please doublecheck?

2

Answer: yes, this is my mistake, I correct it, thank you

Line 156: I have a similar comment to the previous one: the two expressions on the two

sides of the first equal sign are identical.

Answer: yes, again, this is my mistake, I correct it, thank you

Lines 157, 158, 159. What do you mean exactly with “specified form of type (2)”. Any law

expressed in terms of 4-tensors is manifestly invariant under Lorentz transformations.

Continuity equations are just a particular case. Could you please elaborate?

Answer. I agree that my statement seems trivial. It was more to emphasize that the basic laws of physics must concern pairs of quantities and equal their partial derivatives with respect to space and time, and thus be Lorentz invariant. I am elaborating in the text. If the form of the equations confers Lorentz invariance, as is well known, we emphasize here the reciprocal statement.

Line 204: it is at the beginning slightly surprising that you call g a concentration, when at line 198 you wrote that it is the flux of the concentration f. Then it becomes clear that the velocity w of the fluid is assumed to be uniform, constant, and non-dimensional. But I think it would help the reader if you could explain things in a different order.

 Answer: I explain better in the text this somewhat surprising choice

Line 223: Could you state explicitly the equations describing the boundary and initial

conditions applied? In particular, it is not clear to me what condition is applied at the right

boundary.

Answer: I am adding a more explicit formulation of the right condition, by adding for t = 0.

Line 238: It is the first time I ever encounter the term “détentes”. Is there a standard way

of referring to this concept?

Answer: the term “détente” is rare, I admit. The term “rarefaction wave” is more generally found, I correct the text.

Line 238 and 239: you write “We need to impose an optimization criterion that selects the

weak solutions that make physical sense.” Isn’t there a unique solution, once you applied a

specific set of boundary conditions and initial condition? What is the purpose of imposing

an optimization criterion then? In other words: what is the optimization variable

considered? Is it perhaps the time-dependence of g at the left boundary?

Answer You've hit on an important point. Yes, it's an important property of hyperbolic problems when looking for weak solutions in the mathematical sense, i.e., representing true discontinuities in the sense of distribution theory, to present multiple solutions. Solutions are complete spatio-temporal evolutions g(x,t) linking boundary conditions that have been specified (Riemann problem). I am adding a sentence in the text to stress this point, an original point of the paper.

Line 305, 306, 307, 308. I think it would be very helpful if you could include a picture

representing the thermodynamic cycle that you are describing.

Answer There is no cycle as such, but practitioners of hyperbolic problems trace paths in the isotherms connecting the extreme points of the boundary conditions. In the general case, and for isotherms with more complicated shapes than that in this article, the optimal path in the sense of the entropy condition is given by the convex envelope to the isotherm between the extreme points.

I am adding this information at line 285 of the first version (before former figure 6, now 7)

Round 2

Reviewer 1 Report

Comments and Suggestions for Authors

Second referee report of “Finite time thermodynamics, hyperbolicity, Lorentz invariance. Study of an example,” by Bernard Guy, MS entropy-3623795. Submitted to the journal Entropy.

I have read the author’s response to my first referee report, and, unfortunately, still cannot recommend it for publication in Entropy. I have reviewed a number of research papers for Entropy, many making significant contributions. As I stated in my first referee report, I just don’t think that the author has cleared that bar.

In their title, the author has the explicit phrase: “Study of an example.” So let me focus first on that example, found in the third section of the manuscript. As I stated in my first report, this physical example would be very unlikely to have anything even remotely moving at relativistic velocities. Without fast particles, I don’t see how a “relativistic thermodynamics,” the theoretical structure used here, could be tested. Ordinary Newtonian/Galilean type mechanics, perhaps in its diffusion form, should suffice.

Another problem with the author’s example is that it clearly has a preferred reference frame in the form of the rock medium through which the iron atoms move. This is clearly a physically special reference frame, and with it on the scene, I just don’t see how Lorentz invariance could be appropriate. Let me add that the term “Lorentz invariance” is also in the title of the manuscript, so an error on this point, which I fear has been made, would be serious.

Another issue is the meaning of the term “finite time thermodynamics” in the manuscript. The discipline of finite time thermodynamics analyses thermodynamic processes that take place in finite time, with researchers having to determine the thermodynamic path taken from beginning to end that minimizes the dissipated heat. This relates to geometry of thermodynamics, where minimum dissipation relates to the path that minimizes the geometric length taken from start to finish. But I did not see any such process carried out in the manuscript. Instead the author states beginning line 73: “Similarly, thermodynamics in finite time is understood here in its simplest form, based on the very words that define it: finite time, i.e., the study of systems for which a finite evolution time is considered, whether imposed or derived from the solution of the problem. In this context, the question of temperature, although essential in finite-time
thermodynamics, will not be addressed.”

To me this is clearly an admission by the author that a “finite time thermodynamics” analysis will not be done. It appears that the only use of the term “finite time thermodynamics” is as a long winded way of saying that some quantities in the manuscript are finite, perhaps the time. But every paper in thermodynamics deals with processes that are finite in some way, so the use of the term “finite time thermodynamics” adds nothing of any meaning here. Nevertheless, this term is used in the title, and repeated maybe 10 times in the text. This is very misleading to potential readers. This manuscript has absolutely nothing to do with what researchers think of as “finite time thermodynamics.”

In conclusion, in my opinion, this paper is fundamentally flawed in three essential ways. These three ways all appear in the title. So, I cannot recommend this manuscript for publication in Entropy.
